# Patent ductus arteriosus (also non-hemodynamically significant) correlates with poor outcomes in very low birth weight infants. A multicenter cohort study

Elena Chesi[1]☯, Katia Rossi[2]☯, Gina Ancora[3], Cecilia Baraldi[2], Mara Corradi[4], Francesco Di Dio[1], Giorgia Di Fazzio[5], Silvia Galletti[6], Giovanna Mescoli[7], Irene Papa[3], Agostina Solinas[8], Luca Braglia[9], Antonella Di Caprio[10], Riccardo Cuoghi Costantini[11], Francesca Miselli[2,12]*, Alberto Berardi[2], Giancarlo Gargano[1]

1 Neonatal Intensive Care Unit, Department of Obstetrics and Pediatrics, IRCCS di Reggio Emilia, Reggio Emilia, Italy, 2 Neonatal Intensive Care Unit, Women's and Children's Health Department, University Hospital of Modena, Modena, Italy, 3 Neonatal Intensive Care Unit, Infermi Hospital, Rimini, Italy, 4 Neonatal Intensive Care Unit, Women's and Children's Health Department, AOUP, University of Parma, Parma, Italy, 5 Neonatal Intensive Care Unit, ARNAS Garibaldi Hospital, Catania, Italy, 6 Neonatal Intensive Care Unit, Women's and Children's Health Department, University Hospital Sant'Orsola-Malpighi, Bologna, Italy, 7 Neonatal Intensive Care Unit, Women's and Children's Health Department, Maggiore University Hospital, Bologna, Italy, 8 Neonatal Intensive Care Unit, University Hospital S.Anna, Ferrara, Italy, 9 Biostatistician IRCCS of Reggio Emilia, Italy, 10 School of Pediatrics Residency, University of Modena and Reggio Emilia, Modena, Italy, 11 Department of Medical and Surgical Sciences for Mother, Child and Adult, University of Modena and Reggio Emilia, Modena, Italy, 12 PhD Program in Clinical and Experimental Medicine, University of Modena and Reggio Emilia, Modena, Italy

☯ These authors contributed equally to this work.
* 79638@studenti.unimore.it

**Data Availability Statement:** All relevant data are within the manuscript and its Supporting information files.

## Abstract

### Objectives

To standardize the diagnosis of patent ductus arteriosus (PDA) and report its association with adverse neonatal outcomes in very low birth weight infants (VLBW, birth weight < 1500 g).

### Study design

A multicenter prospective observational study was conducted in Emilia Romagna from March 2018 to October 2019. The association between ultrasound grading of PDA and adverse neonatal outcomes was evaluated after correction for gestational age. A diagnosis of hemodynamically significant PDA (hsPDA) was established when the PDA diameter was ≥ 1.6 mm at the pulmonary end with growing or pulsatile flow pattern, and at least 2 of 3 indexes of pulmonary overcirculation and/or systemic hypoperfusion were present.

### Results

218 VLBW infants were included. Among infants treated for PDA closure in the first postnatal week, up to 40% did not have hsPDA on ultrasound, but experienced clinical worsening. The risk of death was 15 times higher among neonates with non-hemodynamically

**Funding:** The author(s) received no specific funding for this work.

**Competing interests:** The authors have declared that no competing interests exist.

**Abbreviations:** BPD, bronchopulmonary dysplasia; GA, gestational age; HsPDA, hemodynamically significant patent ductus arteriosus; IVH, intraventricular hemorrhage; NEC, necrotizing enterocolitis; NICU, neonatal intensive care unit; PDA, patent ductus arteriosus; ROP, retinopathy of prematurity; US, ultrasound; VLBW, very low birth weight, birth weight < 1500 g; wks, weeks.

significant PDA (non-hsPDA) compared to neonates with no PDA. In contrast, the risk of death was similar between neonates with hsPDA and neonates with no PDA. The occurrence of BPD was 6-fold higher among neonates with hsPDA, with no apparent beneficial role of early treatment for PDA closure. The risk of IVH (grade $\geq$ 3) and ROP (grade $\geq$ 3) increased by 8.7-fold and 18-fold, respectively, when both systemic hypoperfusion and pulmonary overcirculation were present in hsPDA.

## Conclusions

The increased risk of mortality in neonates with non-hsPDA underscores the potential inadequacy of criteria for defining hsPDA within the first 3 postnatal days (as they may be adversely affected by other clinically severe factors, i.e. persistent pulmonary hypertension and mechanical ventilation). Parameters such as length, diameter, and morphology may serve as more suitable ultrasound indicators during this period, to be combined with clinical data for individualized management. Additionally, BPD, IVH (grade $\geq$ 3) and ROP (grade $\geq$ 3) are associated with hsPDA. The existence of an optimal timeframe for closing PDA to minimize these adverse neonatal outcomes remains uncertain.

## Introduction

The incidence of patent ductus arteriosus (PDA) is inversely related to gestational age (GA). Approximately half of preterm infants born before 28 weeks' (wks) gestation present with PDA, particularly if ultrasound (US) scans are performed within the first 3 postnatal days [1]. PDA can result in pulmonary overcirculation and systemic hypoperfusion, the clinical significance of which depends on the size of the left-to-right shunt. The increased risks of death, intraventricular hemorrhage (IVH), bronchopulmonary dysplasia (BPD), necrotizing enterocolitis (NEC) and retinopathy of prematurity (ROP) have been historically associated with PDA [2–4]. Delayed PDA treatment may also be associated with a higher risk of pulmonary hemorrhage, and failure to conduct screening echocardiography in extremely preterm infants within the initial 3 postnatal days is associated with elevated in-hospital mortality [5, 6]. In the last decade, therapeutic strategies for PDA have undergone significant changes. In addition to "classical" pharmacological treatments such as indomethacin and ibuprofen, acetaminophen has also been suggested as an effective drug in closing PDA [7]. However, despite the widespread diffusion of PDA treatment, the comorbidities historically attributed to PDA persistence have not actually decreased. New data also show that PDA can close spontaneously, even in extremely low birth weight infants [8]. Moreover, pharmacological treatment may be associated with significant side effects [7]. As a result, divergent perspectives persist, with one faction asserting that PDA treatment plays a role in preventing adverse neonatal outcomes, while another contends that it does not offer such preventive benefits [9, 10]. A conservative approach to managing PDA, encompassing strategies such as fluid restriction, diuretics, and higher positive airway end-expiratory pressure, has gained increasing favor [11, 12].

Studies and data from national registries present disparate rates of PDA-associated outcomes. These differences may be attributed to variations in methods and timing employed for assessing PDA and its treatment [13]. A large number of investigators have analyzed PDA hemodynamic significance in two ways, either by evaluating echocardiographic indexes only [14–16] or using clinical and US composite scores [17–19]. The term "hemodynamically

significant patent ductus arteriosus (hsPDA)" has been increasingly used, and several protocols have been developed in an attempt to define hsPDA by integrating both clinical and US scores [18, 20]. Recent studies were carried out to identify preterm neonates with signs of pulmonary overcirculation or systemic hypoperfusion in order to tailor PDA treatment [14]. Despite these efforts, a consensus on the precise criteria for identifying hsPDA has yet to be reached. Vermont Oxford Network Registry (VON) data from neonatal Intensive Care Units (NICUs) of a northern region of Italy (Emilia-Romagna) revealed significant variability in the rates of diagnosing and treating PDA. To overcome this variability, a protocol was set up by the Emilia-Romagna Study Group with the aim to achieve shared US criteria for hsPDA diagnosis, and to develop a comprehensive PDA scoring system integrating both US and clinical findings. We hypothesized that a consensus on the definition and grading of PDA could enhance management, thereby reducing adverse neonatal outcomes.

The study objectives were 1) to standardize the diagnosis of hsPDA across the seven NICUs in the Emilia-Romagna region, and 2) to investigate associations between PDA severity grading and adverse neonatal outcomes.

## Methods

This was an observational multicenter prospective study, carried out in Emilia-Romagna (Italy) from March 1st, 2018, to October 31st, 2019. Newborns with a birth weight < 1500 g (very low birth weight, VLBW) and/or GA ≤ 29 wks were enrolled. Exclusion criteria were: congenital malformations or first functional echocardiography performed after 72 hours of life. For data collection purposes, a standardized form was utilized to gather relevant data for each enrolled patient (chorioamnionitis, maternal hypertension/eclampsia and gestational diabetes, antenatal steroid use, GA, birth weight, 5-minute Apgar score, mode of delivery and surfactant administration, echocardiograms, treatment for PDA closure, neonatal outcomes). Adverse neonatal outcomes were defined as follows: death, pulmonary hemorrhage, IVH (classified according to Papile) [21], ROP (classified according to the International Classification of Retinopathy of Prematurity) [22], stage ≥ 2 NEC (classified according to Bell's criteria) [23], and BPD (defined as oxygen requirement at 36 weeks corrected age) [24]. Each form for data collection was accessed online through a password protected system. A written informed consent was obtained from the parents of the neonates enrolled in the study. The study was approved by the Ethics Committee of the participating centers (reference 2017/0015900).

### Ultrasound and clinical grading of PDA

Periodic meetings were held among centers to analyze the literature and to share US criteria for diagnosis and treatment of hsPDA. Neonatologists performed functional US studies using 4–12 Hz probes (Phillips S.p.A., Milano) according to international guidelines [25]. The US studies were performed by neonatologists who received specific training in pediatric echocardiography (at the master's level). Two-dimensional, M-mode, pulse and color flow Doppler imaging were performed. The PDA diameter was assessed based on at least two measurements, with the average taken from the two measurements. Hemodynamic evaluation was performed if PDA diameter was ≥ 1.6 mm at the pulmonary end [14, 26], and its flow was growing or pulsatile pattern [15]. PDA was defined as hsPDA if at least 2 of 3 indexes of pulmonary overcirculation and/or systemic hypoperfusion were present (Table 1) [27, 28]. Echocardiographic assessments were repeated during the hospital stay based on clinical judgment. All neonates underwent US before hospital discharge. Table 2 shows the clinical severity grading: clinical criteria were modified from the McNamara scoring system [17]. A composite staging system (ranging from G0 to G3) was developed by combining clinical severity and US grading (Fig 1).

**Table 1. Echocardiographic grading of PDA.** The US grading of PDA involves assessing its severity using the following criteria: PDA diameter, flow characteristics, and indexes of pulmonary overcirculation and/or systemic hypoperfusion.

| Ultrasound score | PDA diameter and flow | | | Indices of pulmonary overcirculation and/or systemic hypoperfusion |
|---|---|---|---|---|
| | PDA d <1.6 mm | PDA d ≥1.6 mm | PDA Flow: Growing or pulsatile | LPAedv > 0.2 m/sec and/or LA/Ao ≥1.5 and/or DAo Flow: absence or reverse |
| E1 | +/- | +/- | - | None present |
| E2 | - | + | + | At least 1 index |
| E3 | - | + | + | At least 2 indexes |
| E4 | - | + | + | All present |

DAo, descending aorta; LPAedv, left pulmonary artery end-diastolic velocity; LA/Ao, left atrial to aortic root ratio; PDA, patent ductus arteriosus.

**Table 2. Grading of the severity of clinical conditions.**

| | NIV | MV | $FiO_2 > 0.28$ | pH<7.1 | BE>-12 | MBP<30 mmHg | AKI | X-ray Cardiomegaly | NEC ≥Bell's stage 2 | Bleeding | Clinical severity |
|---|---|---|---|---|---|---|---|---|---|---|---|
| C1 | - | - | - | - | | - | - | - | - | - | - |
| C2 | + | - | - | - | - | - | - | - | - | - | Mild |
| C3 | +/- | +/- | + | - | - | +/- | - | - | - | - | Moderate |
| C4 | - | + | + | + | + | +/- | +/- | +/- | +/- | +/- | Severe |

AKI, acute kidney injuryº; BE, bases excess; FiO2, oxygen inspiratory fraction; HFNC, high-flow nasal cannula; MBP, mean blood pressure*; MV, invasive mechanical ventilation; nCPAP, nasal continuous positive airway pressure; NEC, neonatal necrotizing enterocolitis; NIV, non-invasive ventilation.

ºAKI defined as serum creatinine>1.5 mg/dl or diuresis <0.5 ml/kg/h. From: Jetton JG, Askenazi DJ. Update on acute kidney injury in the neonate. Curr Opin Pediatr. 2012;24(2):191–6.

*Hypotension defined as MBP<30 mmHg in the first 72 hours of life. From: Dempsey EM, Barrington KJ. Diagnostic criteria and therapeutic interventions for the hypotensive very low birth weight infant. J Perinatol. 2006;26(11):677–81.

### Timing and mode of treatment

The decision to initiate treatment for PDA closure was left to the attending clinician's discretion. The drugs administered for PDA closure included ibuprofen (loading dose 10 mg/kg, maintenance dose 5 mg/kg/day), acetaminophen (15 mg/kg every 6 hours), and indomethacin (loading dose 0.2 mg/kg, maintenance dose 0.1 mg/kg/day). PDA treatment was classified as early (postnatal age ≤ 7 days) or late (postnatal age > 7 days). Surgical PDA closure was performed when medical treatment failed or was contraindicated. All PDA ligations were performed on-site by a mobile cardiosurgical team.

### Adverse neonatal outcomes

Patient records provided the following data related to adverse neonatal outcomes: death, pulmonary hemorrhage, IVH, ROP, stage ≥ 2 NEC, and BPD.

### Data analysis

Statistical analysis was carried out using SPSS software 23.0. Continuous variables were expressed as medians and interquartile ranges (IQR). Categorical data were expressed as numbers and percentages. The Student's t-test was used for unadjusted comparisons of continuous variables between groups; Pearson's $\chi^2$ test and Fisher's exact test (when one or more cells had expected frequency < 5) were used for unadjusted comparisons of categorical variables between groups. To assess a potential association between US PDA scoring and neonatal

| | | ULTRASOUND GRADING (E) | | | | |
|---|---|---|---|---|---|---|
| | | E0 | E1 | E2 | E3 | E4 |
| CLINICAL GRADING (C) | C1 | E0-C1 | E1-C1 | E2- C1 | E3- C1 | E4- C1 |
| | C2 | E0-C2 | E1-C2 | E2- C2 | E3- C2 | E4- C2 |
| | C3 | E0-C3 | E1-C3 | E2- C3 | E3- C3 | E4-C3 |
| | C4 | E0-C4 | E1-C4 | E2- C4 | E3- C4 | E4-C4 |
| | | | | | | |
| COMBINED GRADING (G) | | G0 | G1 | G2 | G3 | |

**Fig 1. Combined clinical and ultrasound grading (G) at 72 hours of life.** G0 (white boxes): infants without PDA (E0) whatever the clinical grading. G1 (pink boxes): infants with PDA (E1) whatever the clinical grading or asymptomatic infants (C1) with PDA. G2 (gray boxes): symptomatic infants (C2-C4) with PDA (E2) or infants with hsPDA (E3-E4) and mild symptoms (C2). G3 (red boxes): infants with hsPDA (E3-E4) and moderate-to-severe clinical conditions (C3-C4).

outcomes, we estimated bivariate logistic regression models correcting for GA. The normality of the residuals of each model was assessed by using the standardized residuals, Q-Q plot. The results were reported as odds ratios (ORs), with 95% confidence interval (CI) and p-values. For all analyses, p-values were considered significant when below the alpha level set at 0.05.

## Results

### Study population

Out of 267 VLBW infants, 49 were ruled out (Fig 2), leaving a total of 218 subjects included in the study. Demographic, maternal, and neonatal characteristics are presented in Table 3, stratified by gestational age (GA < 26 weeks' versus ≥ 26 weeks' gestation).

### US data and composite staging

Among 218 VLBW neonates included in the study, 139 (64%) underwent the first US assessment within 48 hours of life. PDA was more likely to be diagnosed within 48 hours of life (107/139, 77%) than at 49–72 hours of life (38/79, 48%; p<0.001) (Fig 2). During the initial 72 hours of life, a total of 155 out of 218 newborns (71%) underwent one US study, while 63

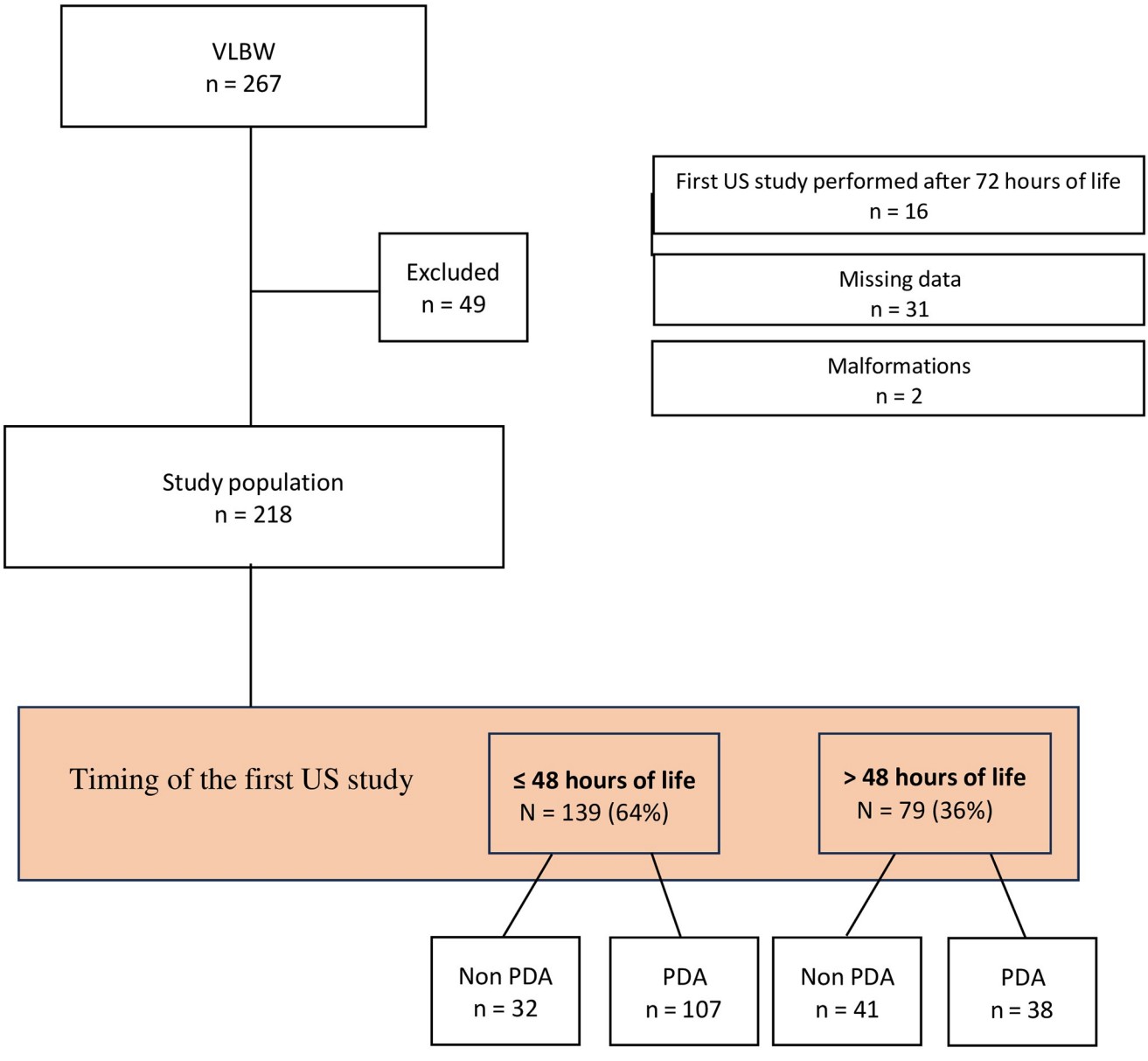

**Fig 2. Flow diagram of the study.**

newborns (29%) underwent two US studies. At 72 hours of life, PDA was present in 127 neonates (58%). Based on the combined clinical and US grading of PDA within 72 hours of life (Fig 1), a total of 91 neonates were classified as G0, 71 neonates were classified as G1, 30 neonates were classified as G2, and 26 neonates were classified as G3.

## PDA treatment

Overall, 70 of 218 neonates (32%) received at least one course of treatment. Treatment for PDA closure was more frequent in neonates with lower GA (28 out of 42 neonates < 26 wks' gestation, 67% vs 42 out of 176 neonates ≥ 26 wks' gestation, 24%, p<0.001).

**Table 3. Demographics and clinical findings of the study population according to gestational age: < 26 weeks' gestation vs ≥ 26 weeks' gestation.**

| | GA< 26 (n = 42) | Missing Data[†] | GA≥ 26 (n = 176) | Missing Data[†] | p |
|---|---|---|---|---|---|
| APGAR 5˚, median [IQR] | 7 [5.8; 8] | 2 | 9 [8; 9] | 1 | 0.024 |
| Complete antenatal steroids, n (%) | 19 (56%) | 8 | 136 (85%) | 15 | <0.001 |
| Cesarean section, n (%) | 22 (54%) | 1 | 140 (81%) | 2 | <0.001 |
| Vaginal delivery, n (%) | 18 (44%) | 1 | 32 (18%) | 2 | <0.001 |
| Chorioamnionitis, n (%) | 20 (49%) | 1 | 36 (21%) | 5 | <0.001 |
| Maternal hypertension/eclampsia n (%) | 4 (10%) | 1 | 36 (21%) | 3 | 0.154 |
| Gestational diabetes, n (%) | 1 (3%) | 2 | 24 (14%) | 2 | 0.074 |
| No surfactant administration, n (%) | 4 (10%) | 2 | 96 (55%) | 2 | <0.001 |
| PDA at 72 hours of life, n (%) | 31 (74%) | - | 96 (55%) | - | 0.035 |
| Early PDA treatment, n (%) | 23 (55%) | - | 38 (22%) | - | <0.001 |
| Late PDA treatment, n (%) | 5 (12%) | - | 4 (2%) | - | 0.017 |
| PDA surgical ligation, n (%) | 9 (21%) | - | 1 (1%) | - | <0.001 |
| Persistent PDA at discharge, n (%) * | 1 (3%) | 2 | 8 (5%) | 3 | 0.840 |
| Death, n (%) | 9 (21%) | - | 3 (2%) | - | <0.001 |

GA, gestational age in weeks. PDA, patent ductus arteriosus.

[†]Percentages and significance are calculated based on the patients who were tested (i.e. after excluding missing cases).

* Percentages are calculated among 33 infants with GA < 26 weeks' gestation and 173 infants with GA ≥ 26 weeks' gestation who survived to discharge, excluding missing data.

**Early treatment.** Among 127 neonates with PDA diagnosed within the first 72 hours of life, 61 (48%) underwent early treatment for PDA closure (i.e. within the first 7 postnatal days; medical treatment, n = 60, surgical treatment, n = 1). Fig 3 shows the number of infants who received early treatment according to the US and clinical composite staging subgroup (ranging from G1 to G3). Among the 61 early treated infants, 38 had an hsPDA (E3 n = 28; E4 n = 10), while 23 had a PDA that was not diagnosed as hemodynamically significant (E1 n = 12; E2 n = 11). All the infants with E1 PDA (no signs of pulmonary overcirculation or systemic hypoperfusion) received early treatment for PDA closure due to moderate-to-severe clinical worsening.

PDA was more likely to be treated early in infants with lower GA (<26 wks' gestation, 23/31, 74% vs ≥ 26 wks' gestation, 38/96, 40%, p = 0.024). Early medical treatment for PDA closure included ibuprofen (n = 46), paracetamol (n = 13) and indomethacin (n = 1). Among infants treated with ibuprofen, two developed oliguria, and one experienced gastric bleeding. The median age at the initiation of early treatment was 2 days, with a median duration of medical treatment lasting 3 days. Among the early medically treated neonates, the PDA closed in

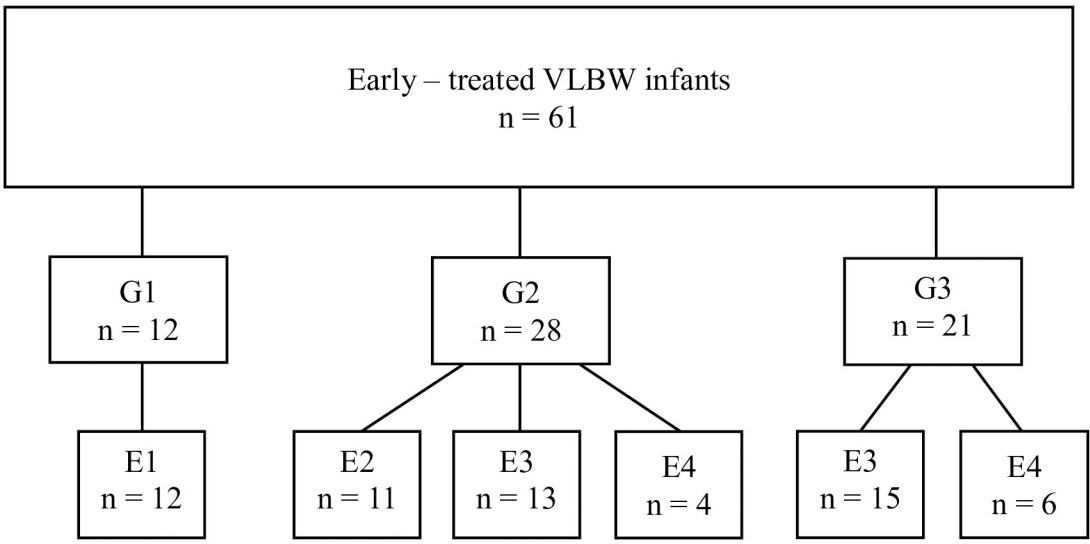

**Fig 3. Combined clinical and US severity score (ranging from G1 to G3) among 61 early-treated infants.** The PDA US severity score is also reported for each group. VLBW, very low birth weight.

42 out of 60 (70%) cases. Thirteen neonates (13/60, 22%) underwent a second treatment, with surgical intervention in 5 neonates and medical intervention in 8 neonates (2 of whom subsequently underwent surgical treatment) (Fig 4).

**Late treatment.** Nine neonates (GA < 26 wks n = 5; GA ≥ 26 wks n = 4) received late treatment for PDA closure (i.e. after the first 7 postnatal days). Of these, 8 newborns received medical treatment (ibuprofen n = 1, paracetamol n = 5 and indomethacin n = 2) and one infant underwent surgical ligation. Among 8 late medically treated infants, the PDA closed in 4 (50%). The median age at initiation of treatment was 13 days, with median duration of medical treatment lasting 3.5 days. None of the infants who were medically treated after the first 7 postnatal days experienced adverse effects.

**Surgical treatment.** Overall, ten infants (10/218, 5%) underwent surgical ligation (median age at treatment 23.6 days of life). Six of ten surgically treated newborns (60%) developed complications (post ligation syndrome n = 4, vocal cord paralysis n = 2); none of them died.

## Outcomes

Among 206 surviving infants, 9 (4%) had persistent PDA at the time of hospital discharge. Twelve infants died (median age 9.5 days), and none within the first 72 hours of life. Mortality was higher in those with GA < 26 wks (p<0.001, Table 3).

Table 4 presents the analysis of adverse neonatal outcomes based on US PDA grading after correction for GA. NEC and pulmonary hemorrhage were excluded from the analysis due to the small number of events. After adjusting for GA, the risk of mortality was higher among infants with E2 PDA. Conversely, the risk of BPD was higher among infants with hsPDA (E3 and E4). Among those who received early treatment, the incidence of BPD was lower among infants who responded to treatment with PDA closure (19 out of 38, 50%) compared to those who failed to respond (12 out of 16, 75%), but this difference did not reach statistical significance (p = 0.133). There was also a trend towards the development of any grade ROP in patients with E2 PDA. Finally, we identified an increased risk of IVH (grade ≥ 3) and ROP (grade ≥ 3) in infants with E4 PDA.

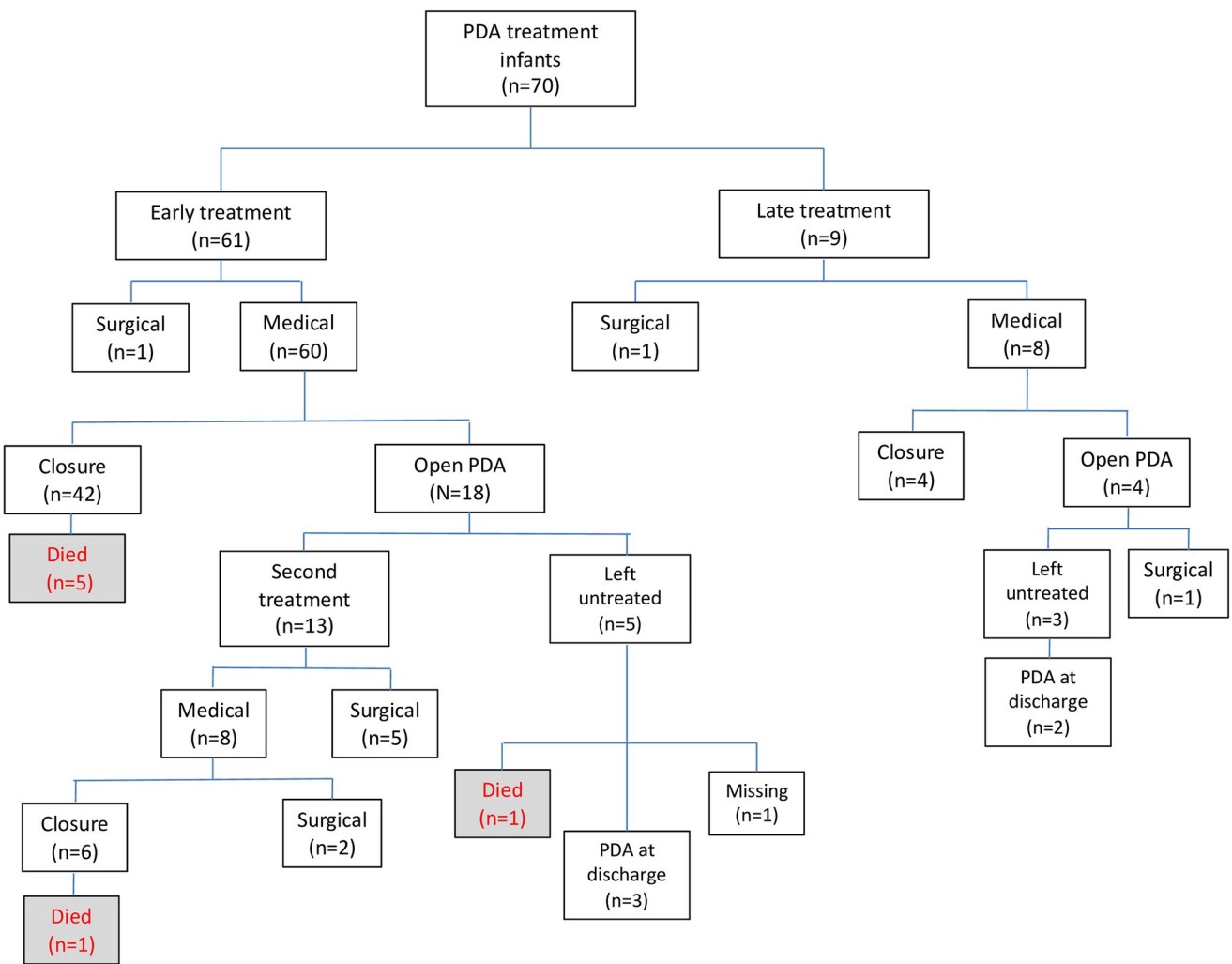

**Fig 4. Flow diagram of patients treated for PDA closure.** A flow diagram illustrating infants treated for PDA closure, including both early and/or late medical and/or surgical interventions, is presented. Early PDA treatment: any treatment in the first seven postnatal days. Late PDA treatment: any treatment after the first seven postnatal days.

## Discussion

Our study shows that approximately 26% of neonates <26 wks' gestation experience a spontaneous PDA closure, consistent with findings in the previous literature [8, 29]. About half of the enrolled infants with evidence of PDA on US received medical treatment: consistently with findings from previous studies, the success rate was about 70% [11, 12, 30–32]. By monitoring neonates during their entire hospitalization, we observed that about 4% had persistent PDA at the time of discharge. This proportion is lower than the rate reported by Clymann and Borràs-Novell, which was approximately 25% [29, 32]. In our cohort, no newborn underwent surgical ligation of PDA after hospital discharge. This is in line with previous studies, suggesting that a high percentage of persistent PDA at discharge in VLBW infants may subsequently close spontaneously (up to 86%) [33].

Consistent with the findings of a national survey [34], ibuprofen was administered as first-line drug for early treatment in 75% of cases. The rates of surgical ligation (14%) and its related

**Table 4. Correlation of adverse neonatal outcomes and ultrasound PDA grading (E0 to E4) after correction for gestational age.**

| OUTCOMES | Ultrasound PDA grading | OR | 95% CI | | p | N |
|---|---|---|---|---|---|---|
| **Death** | E0 | Ref | | | | 218 |
| **Death** | E1 | 7.63 | (0.70–83.32) | | 0.096 | |
| **Death** | E2 | 14.11 | 1.25 | 158.58 | **0.032** | |
| **Death** | E3 | 2.62 | 0.21 | 32.69 | 0.455 | |
| **Death** | E4 | 11.86 | 0.55 | 256.28 | 0.115 | |
| *Death* | *GA* | *0.49* | *0.33* | *0.73* | *0.001* | |
| **BPD** | E0 | Ref | | | | 207* |
| **BPD** | E1 | 1.98 | 0.70 | 5.62 | 0.197 | |
| **BPD** | E2 | 1.47 | 0.33 | 6.54 | 0.615 | |
| **BPD** | E3 | 6.75 | 1.88 | 24.26 | **0.003** | |
| **BPD** | E4 | 6.46 | 1.36 | 30.73 | **0.019** | |
| *BPD* | *GA* | *0.49* | *0.39* | *0.61* | *0.000* | |
| **IVH ≥ 3** | E0 | Ref | | | | 216** |
| **IVH ≥ 3** | E1 | 1.24 | 0.24 | 6.50 | 0.795 | |
| **IVH ≥ 3** | E2 | 2.11 | 0.31 | 14.25 | 0.443 | |
| **IVH ≥ 3** | E3 | 2.34 | 0.46 | 12.01 | 0.308 | |
| **IVH ≥ 3** | E4 | 8.74 | 1.47 | 52.07 | **0.017** | |
| *IVH ≥ 3* | *GA* | *0.75* | *0.58* | *0.97* | *0.026* | |
| **ROP** | E0 | Ref | | | | 208*** |
| **ROP** | E1 | 1.47 | 0.43 | 5.05 | 0.538 | |
| **ROP** | E2 | 4.83 | 1.00 | 23.29 | **0.050** | |
| **ROP** | E3 | 0.52 | 0.13 | 2.08 | 0.355 | |
| **ROP** | E4 | 6.76 | 1.23 | 37.12 | **0.028** | |
| **ROP** | GA | 0.42 | 0.31 | 0.55 | 0.000 | |
| **ROP ≥ 3** | E0 | Ref | | | | 208*** |
| **ROP ≥ 3** | E1 | 1.07 | 0.19 | 5.93 | 0.940 | |
| **ROP ≥ 3** | E2 | 3.21 | 0.51 | 20.42 | 0.216 | |
| **ROP ≥ 3** | E3 | 1.62 | 0.34 | 7.70 | 0.547 | |
| **ROP ≥ 3** | E4 | 17.97 | 2.34 | 137.93 | **0.005** | |
| *ROP ≥ 3* | *GA* | *0.49* | *0.35* | *0.69* | *0.000* | |

BPD, bronchopulmonary dysplasia; GA, gestational age; IVH, intraventricular hemorrhage; PDA, patent ductus arteriosus: ROP, retinopathy of prematurity; US, ultrasound.

\* Among 207 surviving infants at 36 weeks' gestation (eleven patients were excluded because they died before 36 weeks' gestation).

\*\* Among 216 patients, 2 missing data.

\*\*\* Among 208 surviving infants at 32 weeks' gestation (ten patients were excluded because they died before 32 weeks' gestation).

complications were similar to those previously described in the literature [11, 12, 29, 35–37]. However, we observed a higher incidence of post-ligation syndrome (up to 60%), nearly double that reported by Moin et al [38]. This discrepancy may be due to our definition of post-ligation syndrome, which we did not define as the need for vasopressors within 72 hours post-procedure, but rather as a reduction in cardiac output on US within 6 hours of the procedure, possibly encompassing mild cases. In fact, no patient died from post-ligation syndrome in our cohort. Notably, percutaneous closure has demonstrated efficacy in PDA closure with several potential benefits over surgical ligation, as it eliminates the need for incisions and mitigates issues related to manipulating the lung for PDA access, along with reducing the risk of complications such as laryngeal recurrent nerve damage. These benefits contribute to the appeal of

percutaneous closure as a favorable alternative, and this will be the focus of the PIVOTAL study [39].

Although it is commonly recommended to treat only patients with hsPDA based on both US and clinical deterioration, our data reveal the predominant influence of clinical conditions over US findings in the decision to treat neonates for PDA closure. In fact, up to approximately 40% of early treated infants had experienced moderate-to-severe clinical worsening even in the absence of evidence of hsPDA on US. This "overtreatment" could be related to clinicians' inclination to consider the presence of the PDA in unstable infants as an aggravating factor. It is difficult to distinguish the role of the PDA itself versus other factors (preterm birth, perinatal transition, maternal chorioamnionitis, and hypertension) in determining the patient's hemodynamic instability [40]. In our study, the risk of death was nearly 15-fold higher among neonates with E2 PDA (non-hsPDA), as compared to those without PDA (E0), independently from the GA. We highlight this higher risk of death observed in newborns with PDA who do not exhibit US findings of hsPDA, which merits further consideration. Several factors may explain this increased risk of death. First, the standard recognized US criteria for defining an hsPDA may be inadequate for this temporal window (within 72 hours of birth). In fact, persistent pulmonary hypertension of the newborn and/or high pulmonary pressures due to mechanical ventilation may affect the left-to-right shunt through the PDA, temporarily minimizing the pulmonary overcirculation. Also, the typical vasoconstriction of the fetus and an "inappropriate" fluid intake may contribute to a more prolonged equilibrium between systemic and pulmonary pressures. Finally, the use of inotropes and vasopressors in critically ill newborns may reduce the clinical and US findings of systemic hypoperfusion. Thus, from a clinical perspective, it appears that the most appropriate criteria for defining an hsPDA in the early hours of life may be the length, diameter and morphology of the PDA. Indeed, these factors are assumed to be the main determinants for subsequent pulmonary overcirculation and systemic hypoperfusion [41]. Given that PDA diameter in isolation is known to be poorly predictive, it may be beneficial to combine it with clinical data for a more comprehensive grading [42].

Surprisingly, the risk of death adjusted for GA in neonates with hsPDA (E3-E4) was similar to that of neonates without PDA (E0) in our population. One possible explanation is that almost all infants with hsPDA (E3-E4) receive early treatment to close the PDA; this is not the case for infants with non-hsPDA (E2), for whom clinicians are more prone to "watchful waiting". Consequently, the proportion of treatment in E2 patients is lower, and the PDA may remain open, potentially leading to higher mortality. It is important to acknowledge that a significant limitation of the association between non-hsPDA and death in our cohort is the small number of deceased infants. The small sample size of deceased individuals limits the statistical power and generalizability of our findings.

In our study, we observed a 6-fold higher occurrence of BPD in hsPDA infants (E3-4) compared to newborns without PDA (E0), independent of GA. Indeed, previous epidemiological studies highlighted a strong association between PDA and BPD [43], although a causal relationship has not been proven. It is possible that the pulmonary overcirculation, common in infants with hsPDA, plays a role in promoting BPD development. In our cohort, almost all the newborns with hsPDA received treatment for PDA closure, whether medical or surgical. However, PDA closure after early treatment did not appear to significantly reduce the risk of developing BPD. This finding contrasts with previous studies that demonstrated a reduced BPD incidence following more aggressive treatment approaches in the first days of life [43], before the PDA caused clinical symptoms related to its hemodynamics [44]. This discrepancy could be related to our small sample size of neonates with BPD, which may have limited the statistical power to detect the protective role of early PDA closure.

In our study, the risk of IVH (grade $\geq$ 3) and ROP (grade $\geq$ 3) was 8.7-fold and 18-fold higher, respectively, when hsPDA was most significant (E4) as compared to newborns without PDA (E0), independently from GA. Notably, systemic hypoperfusion (always present as a defining criterion for E4 staging) may be a risk factor for both IVH and ROP, and may favor their occurrence. Interestingly, we observed a trend towards higher risk of ROP (all stages included) among infants with non-hsPDA (E2) who survived to 32 wks' gestation.

## Conclusions

Despite standardization of US criteria for PDA diagnosis, clinical conditions still inform up to 50% of the decision to early treat PDA. Surprisingly, after adjusting for GA, the risk of death is similar among infants with hsPDA and infants without PDA. In contrast, the risk of death is nearly 15-fold higher among neonates with non hsPDA. We highlight the increased risk of death in newborns with PDA who do not exhibit US findings of hsPDA, suggesting that the standard recognized US criteria for defining hsPDA may be inadequate within the first 3 post-natal days. Instead, length, diameter and morphology of the PDA may be more appropriate parameters, to be integrated with clinical data for combined grading and an individualized approach. Interestingly, among infants with non-hsPDA, the risk of developing ROP (all stages included) is also higher.

Infants with hsPDA show 6-fold higher risk of BPD compared to infants with no PDA, independently from GA. Finally, the risk of grade $\geq$ 3 IVH and grade $\geq$ 3 ROP is 8.7-fold and 18-fold higher, respectively, when hsPDA is most significant with all signs of systemic hypo-perfusion, compared to newborns without PDA, independently from GA.

Further investigation is needed to better clarify whether an optimal window for PDA evaluation and closure exists, helping to reduce the incidence of death and preterm associated comorbidities, including BPD, IVH and ROP. To date, it is unclear which criteria of hsPDA are most accurate in different temporal windows, and whether there exists an optimal window for PDA evaluation and closure.

## Supporting information

**S1 Data.**
(XLSX)

## Acknowledgments

We would like to thank Daniela Masi (Azienda USL-IRCCS di Reggio Emilia) for English editing. Special thanks to Dr. Cristina Cicero and Dr. Maria Laura Malaigia.

## Author Contributions

**Conceptualization:** Elena Chesi, Katia Rossi, Francesca Miselli, Alberto Berardi, Giancarlo Gargano.

**Data curation:** Elena Chesi, Katia Rossi, Luca Braglia, Francesca Miselli, Alberto Berardi, Giancarlo Gargano.

**Formal analysis:** Luca Braglia.

**Investigation:** Elena Chesi, Katia Rossi, Antonella Di Caprio, Francesca Miselli, Alberto Berardi.

**Methodology:** Elena Chesi, Katia Rossi, Francesca Miselli.

**Project administration:** Elena Chesi, Katia Rossi, Francesca Miselli.

**Resources:** Elena Chesi, Katia Rossi, Antonella Di Caprio, Francesca Miselli.

**Supervision:** Elena Chesi, Katia Rossi, Gina Ancora, Cecilia Baraldi, Mara Corradi, Francesco Di Dio, Giorgia Di Fazzio, Silvia Galletti, Giovanna Mescoli, Irene Papa, Agostina Solinas, Antonella Di Caprio, Riccardo Cuoghi Costantini, Francesca Miselli, Alberto Berardi, Giancarlo Gargano.

**Validation:** Elena Chesi, Katia Rossi, Gina Ancora, Cecilia Baraldi, Mara Corradi, Francesco Di Dio, Giorgia Di Fazzio, Silvia Galletti, Giovanna Mescoli, Irene Papa, Agostina Solinas, Antonella Di Caprio, Riccardo Cuoghi Costantini, Francesca Miselli, Alberto Berardi, Giancarlo Gargano.

**Visualization:** Elena Chesi, Katia Rossi, Cecilia Baraldi, Francesco Di Dio, Giorgia Di Fazzio, Silvia Galletti, Giovanna Mescoli, Irene Papa, Agostina Solinas, Antonella Di Caprio, Riccardo Cuoghi Costantini, Francesca Miselli, Alberto Berardi, Giancarlo Gargano.

**Writing – original draft:** Elena Chesi, Katia Rossi, Francesca Miselli.

**Writing – review & editing:** Elena Chesi, Katia Rossi, Antonella Di Caprio, Francesca Miselli, Alberto Berardi, Giancarlo Gargano.

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
