## [Decision Letter · Decision Letter 0]

22 Nov 2023

PONE-D-23-29459Patent ductus arteriosus: diagnosis, management and adverse neonatal outcomes in preterm Very Low Birth Weight Infants. Area-based prospective multicenter observational study.PLOS ONE

Dear Dr. Miselli,

Thank you for submitting your manuscript to PLOS ONE. After careful consideration, we feel that it has merit but does not fully meet PLOS ONE’s publication criteria as it currently stands. Therefore, we invite you to submit a revised version of the manuscript that addresses the points raised during the review process.

We look forward to receiving your revised manuscript.

Kind regards,

Hakan Aylanc

Academic Editor

PLOS ONE

Journal Requirements:

Reviewers' comments:

Reviewer's Responses to Questions

**Comments to the Author**

1. Is the manuscript technically sound, and do the data support the conclusions?

Reviewer #1: Yes

Reviewer #2: Yes

2. Has the statistical analysis been performed appropriately and rigorously? 

Reviewer #1: Yes

Reviewer #2: I Don't Know

3. Have the authors made all data underlying the findings in their manuscript fully available?

Reviewer #1: Yes

Reviewer #2: Yes

4. Is the manuscript presented in an intelligible fashion and written in standard English?

Reviewer #1: No

Reviewer #2: Yes

5. Review Comments to the Author

Reviewer #1: Dear Authors,

I evaluated the study named “Patent ductus arteriosus: diagnosis, management and adverse neonatal outcomes in preterm Very Low Birth Weight Infants. Area-based prospective multicenter observational study”, with great pleasure and interest. This is a relevant study. However, the manuscript may be suitable for publication after the correction is made. I listed my recommendations and comments below

Overall assessment

In this study, the authors found that when clinical and echocardiographic collaboration were used in the treatment of PDA, about 40 % of patients were treated as medical due to deterioration of the clinic in asymptomatic patients with PDA. Prevalence of death, intraventricular hemorrhage, retinopathy of prematurity, and bronchopulmonary dysplasia were higher in patients with PDA

Patent ductus arteriosus is one of the most common neonatal morbidity in neonatal intensive care but the relationship between the ductus arteriosus and neonatal morbidity remains unclear despite many research papers. Echocardiography is the gold standard for identifying significant PDA. However, every hs-PDA with echocardiographic findings may not cause significant clinical effects. Besides, there is still ongoing debate about the treatment approaches such as conservative? medical treatment? observation?. Therefore, the approaches vary among neonatal intensive care units. In this context, The study is much more important in terms of emphasizing the significance of the individualized approach in the current treatment of PDA

1. The title is not exactly reflective of the study or its content. could Authors please write a more appropriate and clear title to attract readers and encompass your study

The abstract conclusion should involve summarising the key points of the study and presenting the implications of its findings. The conclusion could be revised to get a clear understanding and should be far more specific to the study.

Introduction

The hypothesis of the article is not clear. Authors should determine a more concise, testable, and specific statement of the research question. The hypothesis should directly support the main research question. This should establish a clear link between the hypoyhesis and the purpose of the study

Methodology

Line 113 (An observational prospective area-based study…) may be more appropriate if changed as a multicenter observational study

Authors don't have a control group, why did the authors consider having a control group that was evaluated with echocardiography performed by a cardiologist?

Line 114: authors should not write the aim of the study in the method section

Line 118 For data collection purposes, standardized form… what was the component standardized form used by the authors? The authors should elaborate on its content, providing more details and explanations

Ultrasound and clinical grading of PDA

The authors should provide some insight into the reliability of the neonatologist who conducted the functional echocardiography. For example, how or how many times was PDA measured (one or more), which can be influenced by straining, crying, and cold environment, Which measurement was confirmed PDA? single or the average of measurements?

Please authors should give definitions of AKI, hypotension, and other morbidities such as NEC, BPD, and ROP in the method section. Authors should add these definitions and references to the methods

Short-term clinical outcomes

Authors should state other morbidities related to PDA such as respiratory support (CPAP and mechanical ventilation), surfactant administration, surfactant doses, extubation failure, postnatal steroids…

Line 147: Data analysis Statistical analysis was carried out using SPSS software 23.0. … which test was used to assess the normality distribution of data. The appropriate statistical method should be written for each specific situation

Line 147: Continuous variables were expressed as the means ± standard deviations or medians and interquartile ranges. Authors presented all continuous data as median and interquartile so you don’t need to use means and standard deviation

Figure 1 is very interesting. Authors should add Figure 1 to the methods section, explaining how to make or comment because it will guide readers and enhance the overall understanding of the figure’s significance in the context of the research

Line 152: “Confidence Interval (CI) was 0.95; when p < 0.05, tests were considered

statistically significant” should be the final sentence for the data analysis section.

Line 154: ‘The Strengthening the Reporting of Observational Studies in Epidemiology (STROBE) reporting guidelines were followed for this study’ The sentence seems unnecessary because authors must follow it in observational studies to structure their manuscript, making it easier to be understood

Line 178: Demographic and further maternal and neonatal characteristics according to gestational age (GA < 26 versus . 26 wks gestation) are shown in Table 3. This sentence might have been changed to “Demographic, maternal and neonatal characteristics are shown in Table 3.

The English language must be revised throughout the text. If the manuscript is checked by a Language Editing in terms of grammar and punctuation, it would be better

In Table 3, there is no need to give as a column in table 3 because the missing data is exclusion criteria

Line 236 ‘The Fisher exact test was used to calculate the p-value for each variable except the Apgar score for which the Kruskal-Wallis¥ test is applied’ should be placed in the statistical section

this sentence need not be repeated under Table 3

Line 242 Among 207 surviving infants at 36 weeks’ gestation. Eleven patients were excluded because they…. please

Please put a comma instead of a dot before eleven patients.

There is the same error in Line 245, which must be corrected.

Line 343 Figure 2 Flow diagram of the study: ultrasound studies and findings.

there is no need to write “Ultrasound studies and findings “

Figure 4:

Flow diagram of patients treated for PDA. Early and/or late medical and/or surgical

treatment are reported. of PDA. PDA: patent ductus arteriosus. Early PDA treatment: any treatment in the first week of life. Late PDA treatment: any treatment after the first week of life.

Please revise to get a clear understanding. You may change as “Flow diagram of patients treated with medical or surgical ligation”

Discussion

The authors should begin the discussion by summarising the main findings of their study concisely. For example, our study shows….

Authors should make much more comprehensive explanations about the clinical implications of their findings. How might these results impact the monitoring and treatment guidelines of PDA?

Please ensure that clear messages will be accessible to broad readers related to your study

Conclusion

Please focus on summarising key findings without unnecessary details

Reviewer #2: Thank you for the opportunity to review this work. I believe the authors efforts merit publication; however, I have added some areas of clarification which I think would strengthen the present manuscript.

Methods:

Chesi and colleagues performed a ~10year prospective, multicenter observational study with the objective of standardizing PDA diagnosis and report associated outcomes for (218) VLBW infants.

Of these, 40% were treated for PDA in the first postnatal week didn’t have HsPDA but exhibited “clinical worsening.”

After adjustment for gestational age, finding include:

Risk of death 15-fold higher among these infants (no US evidence of HsPDA, only clinical suspicion).

Risk of death similar between infants with HsPDA and those without PDA.

I assume that adjustment for GA was an attempt to control for problems of prematurity to address associations between PDA treatment (contributing to or diminishing these outcomes: death, ROP, IVH, BPD etc.). Given that most (>80%) of infants who undergo definitive PDA closure require mechanical ventilation (invasive or non-invasive) I would like to see more data (aside from FiO2) in regard to level and duration of respiratory support as these data are competing risks/influenced the observed differences for mortality (15-fold), BPD (6-fold higher) and ROP (18-fold) among infants with HsPDA, without an observed benefit of early closure.

Please include n (%) for invasive/noninvasive respiratory support and high-frequency ventilation utilization to the clinical finding/demographics table. Further, please provide duration of mechanical ventilation for comparison between groups (PDA closed vs. those without PDA).

Discussion line 283ish: Please add to discussion that while PDA diameter has historically been incorporated into these scores, it is a poor indicator of adverse outcomes in isolation. https://www.sciencedirect.com/science/article/abs/pii/S0378378223001287?via%3Dihub

290: Indeed, the large challenge for treatment of PDA is agreement on what constitutes a hemodynamically significant PDA and is it pathologic as noted for group E4-flow reversal in descending Ao. Conversely, the watch and wait E2 may be subject to protracted mechanical ventilation and related morbidity/mortality. As such, continued work is needed to elucidate when the “watch” ends and what it the optimal method of definitive closure (TCPC or ligation) based on infant criteria (weight, PDA size, morphology, length, presence of absence of infection etc.)

293: Authors appropriately acknowledge the statistical associations were likely impacted by small sample size and low number of deaths (events).

Please include within your discussion that percutaneous closure has demonstrated efficacy for closing HsPDA with several potential benefits over surgical ligation (mainly avoidance of incision and related problems related to manipulating the lung to access PDA or laryngeal recurrent nerve damage). Moreover, it’s worth noting that the PIVOTAL study may help answer some of these questions. https://www.pivotalstudy.org

Conclusion: I agree with the authors that there are knowledge gaps insofar as criteria, timing, and method of assessing for a HsPDA (including concensus definition) and type of definitive closure that minimize adverse outcomes in this population.

6. PLOS authors have the option to publish the peer review history of their article (what does this mean?). If published, this will include your full peer review and any attached files.

Reviewer #1: **Yes: **Associate Professor Fatih BOLAT, MD

Reviewer #2: No

---

## [Author Response · Author response to Decision Letter 0]

4 Jan 2024

Please find a point-by-point response attached.

---

## [Decision Letter · Decision Letter 1]

24 Jan 2024

PONE-D-23-29459R1Patent ductus arteriosus is associated with higher mortality and adverse neonatal outcomes in preterm Very Low Birth Weight Infants. Multicenter prospective area-based observational study.PLOS ONE

Dear Dr. Miselli,

Thank you for submitting your manuscript to PLOS ONE. After careful consideration, we feel that it has merit but does not fully meet PLOS ONE’s publication criteria as it currently stands. Therefore, we invite you to submit a revised version of the manuscript that addresses the points raised during the review process.

We look forward to receiving your revised manuscript.

Kind regards,

Hakan Aylanc

Academic Editor

PLOS ONE

Journal Requirements:

Reviewers' comments:

Reviewer's Responses to Questions

**Comments to the Author**

1. If the authors have adequately addressed your comments raised in a previous round of review and you feel that this manuscript is now acceptable for publication, you may indicate that here to bypass the “Comments to the Author” section, enter your conflict of interest statement in the “Confidential to Editor” section, and submit your "Accept" recommendation.

Reviewer #1: All comments have been addressed

2. Is the manuscript technically sound, and do the data support the conclusions?

Reviewer #1: Partly

3. Has the statistical analysis been performed appropriately and rigorously? 

Reviewer #1: Yes

4. Have the authors made all data underlying the findings in their manuscript fully available?

Reviewer #1: Yes

5. Is the manuscript presented in an intelligible fashion and written in standard English?

Reviewer #1: Yes

6. Review Comments to the Author

Reviewer #1: The title still does not exactly reflect the study content. Please write a more appropriate and clear title that is associated with your study

Abstract

Line 52: …Patent Ductus Arteriosus (patent ductus arteriosus (PDA) should be written as lowercase letter

Line 53:… in preterm Very Low Birth Weight Infants (VLBW, birth weight < 1500 g)

Int

it would be more appropriate if changed as `? in very low birth weight infants’

Line 57:The association between ultrasound grading of PDA and adverse neonatal outcomes was evaluated after correction for gestational age (GA)…. you could delete GA because you only use it once in the abstract

Line 62:…. in the first postnatal week, up to 40% did not have hsPDA at ultrasound (US), but… US" should not be abbreviated as it was not mentioned previously.

Line 72: ‘The occurrence of BPD was 6-fold higher among neonates with hsPDA, with no apparent beneficial role of early PDA treatment. The risk of IVH (grade ≥ 3) and ROP (grade ≥ 3) were 8-fold a…..’ Please consider providing exact percentages or numerical values when referring to increased risk

Line 67: In the result section of the abstract, there should be no comment. The authors may add the comments to the conclusion section of the abstract. The conclusion of the abstract should be written as a subheading apart from the result section.

Introduction

There are some typographical and grammatical errors like these:

Line 102: Many investigators analyzed the hemodynamic significance of PDA, either by evaluating only echocardiographic indexes [14-16], or by using both clinical and US composite scores [17-19].

Line 104: “The term “hemodynamically significant patent ductus arteriosus 104 (hsPDA)” has being been increasingly used:..”plesase delete "being"

The language is generally clear, but some sentences could be more concise. Please consider a thorough proofread for grammatical accuracy and clarity.

METHODS

Line 183:…(loading and maintenance dose 10 and 5 mg/kg/die….maintenance dose 0.2 and 0.1 mg/kg/die…. These sentences need to be changed to a clearer definition, die converting to day (to be more understandable for the reader)

RESULT

Complete antenatal steroides (correct it as ‘antenatal steroids’) and write

Table 3:

Those percentage values need to be corrected: For example, 19/42 is 45.2% but it is written as 56% in Table 3.

Cesarean section, Vaginal delivery, Chorioamnionitis, Maternal hypertension/eclampsia, Gestational diabetes…The numbers seem to have been calculated incorrectly.

Mistakes should be corrected throughout the text, tables, and figures.

You should standardize all the numbers throughout the paper and either round them down or round them up to one decimal place (some are rounded up, some are in decimals, some are whole numbers)

Line 228: Overall, 70 of 218 neonates (32.1%).. please standardize the numbers:. … not 32.1% but 32%. All of them need to be in the same numerical type.

Lin 237: no signs of 237 pulmonary over circulation nor systemic hypoperfusion—could you change it as ‘no sign of pulmonary over circulation or systemic hypoperfusion’ or use ‘neither pulmonary over circulation nor systemic hypoperfusion’

Line 245: The total number of patients treated is not 61 patients (when I sum up, it is 60 patients)

Please check all of the numbers and percentages thoroughly again

Again, the numbers in Figure 2 should be standardized as previously described.

Line 261:’ …with median duration of medical treatment lasting 3.5 days.’ How did the median duration of medical treatment last 3.5 days? Didn't you give a 3-day cure treatment?

Line 263: please write the number without decimals.

Discussion

Line 308: In the study, the authors found that post-ligation syndrome was twice as high

How do you explain this situation?

7. PLOS authors have the option to publish the peer review history of their article (what does this mean?). If published, this will include your full peer review and any attached files.

Reviewer #1: **Yes: **Fatih Bolat, MD Associate Professor

---

## [Author Response · Author response to Decision Letter 1]

7 Mar 2024

Please see the attached file (Response to Reviewers)

---

## [Decision Letter · Decision Letter 2]

2 May 2024

PONE-D-23-29459R2Patent Ductus Arteriosus (even non-Hemodynamically Significant) correlates with poor outcomes in Very Low Birth Weight Infants. Multicenter cohort study.PLOS ONE

Dear Dr. Miselli,

Thank you for submitting your manuscript to PLOS ONE. After careful consideration, we feel that it has merit but does not fully meet PLOS ONE’s publication criteria as it currently stands. Therefore, we invite you to submit a revised version of the manuscript that addresses the points raised during the review process.

We look forward to receiving your revised manuscript.

Kind regards,

Hakan Aylanc

Academic Editor

PLOS ONE

Journal Requirements:

Additional Editor Comments:

Dear Author,

The English version of the article needs to be edited. Grammatical errors need to be eliminated. You are expected to correct numerical errors in the tables within the article and respond adequately to the revision request. I will be waiting for your revision. I wish conveniences.

Reviewers' comments:

Reviewer's Responses to Questions

**Comments to the Author**

1. If the authors have adequately addressed your comments raised in a previous round of review and you feel that this manuscript is now acceptable for publication, you may indicate that here to bypass the “Comments to the Author” section, enter your conflict of interest statement in the “Confidential to Editor” section, and submit your "Accept" recommendation.

Reviewer #1: All comments have been addressed

2. Is the manuscript technically sound, and do the data support the conclusions?

Reviewer #1: Yes

3. Has the statistical analysis been performed appropriately and rigorously? 

Reviewer #1: Yes

4. Have the authors made all data underlying the findings in their manuscript fully available?

Reviewer #1: Yes

5. Is the manuscript presented in an intelligible fashion and written in standard English?

Reviewer #1: Yes

6. Review Comments to the Author

Reviewer #1: Dear Editor

I'd like to suggest the manuscript titled "Patent Ductus Arteriosus (even non-Hemodynamically Significant) correlates with poor outcomes in Very Low Birth Weight Infants. Multicenter cohort study" that was submitted to PLOS ONE for review. After reading through the manuscript carefully, I would like to thank the authors' efforts in addressing some of the previous feedback. However, there are still some significant issues that need to be resolved before the manuscript can be considered for publication.

Even after trying to revise the manuscript, there are still quite a few grammar errors throughout the text. These mistakes can make it difficult for readers to understand the content clearly,

SOME SUGGESTIONS:

1. Please condense the conclusion in the abstract, as it is currently too long to provide a concise conclusion

2. Line 55: please change as “Multicenter prospective observational study was conducted in Emilia Romagna from”

3. Line 58:.” A diagnosis of hemodynamically significant (hsPDA) was established when the PDA diameter was ≥ 1.6 mm at the pulmonary end with growing or pulsatile flow, and at least 2 of 3 indexes of pulmonary overcirculation and/or systemic hypoperfusion were present” should be added to study design

3) LİNE 83: PDA can cause result in pulmonary overcirculation and systemic… Please correct

4. Line 103... PDA by evaluating either only…grammar error

4) please calculate the ratio accurately, especially in Table 3

The suggested corrections have not been made in Table 3 “Those percentage values need to be corrected: For example, 19/42 is 45.2% but it is written as 56% in Table 3. Cesarean section, Vaginal delivery, Chorioamnionitis, Maternal hypertension/eclampsia, Gestational diabetes…The numbers seem to have been calculated incorrectly.

4) The manuscript has grammatical errors that should be corrected. For example. Line 55

…. Multicenter prospective observational study was conducted in Emilia Romagna from 56 March 2018 to October 2019.

Please comprehensively check by a native speaker

7. PLOS authors have the option to publish the peer review history of their article (what does this mean?). If published, this will include your full peer review and any attached files.

Reviewer #1: **Yes: **Fatih Bolat, Associate Professor

---

## [Author Response · Author response to Decision Letter 2]

21 Jun 2024

Please see file attached in the "attached files" section.

---

## [Editor Report · Decision Letter 3]

25 Jun 2024

Patent ductus arteriosus (also non-hemodynamically significant) correlates with poor outcomes in very low birth weight infants. A multicenter cohort study

PONE-D-23-29459R3

Dear Dr. Miselli,

We’re pleased to inform you that your manuscript has been judged scientifically suitable for publication and will be formally accepted for publication once it meets all outstanding technical requirements.

Kind regards,

Hakan Aylanc

Academic Editor

PLOS ONE
---

## [Editor Report · Acceptance letter]

28 Jun 2024

PONE-D-23-29459R3 

PLOS ONE

Dear Dr. Miselli, 

I'm pleased to inform you that your manuscript has been deemed suitable for publication in PLOS ONE. Congratulations! Your manuscript is now being handed over to our production team.

Kind regards, 

on behalf of

Dr. Hakan Aylanc 

Academic Editor

PLOS ONE